# Trans-Cinnamaldehyde—Fighting *Streptococcus mutans* Using Nature

**DOI:** 10.3390/pharmaceutics16010113

**Published:** 2024-01-15

**Authors:** Zilefac Brian Ngokwe, Amit Wolfoviz-Zilberman, Esi Sharon, Asher Zabrovsky, Nurit Beyth, Yael Houri-Haddad, Dana Kesler-Shvero

**Affiliations:** 1Department of Prosthodontics, Hadassah Medical Center, Faculty of Dental Medicine, Hebrew University of Jerusalem, Jerusalem 9112001, Israel; brian.ngokwe@mail.huji.ac.il (Z.B.N.); amit.wolfoviz@mail.huji.ac.il (A.W.-Z.); esisharon@hadassah.org.il (E.S.); asherz@hadassah.org.il (A.Z.); nurit.beyth@mail.huji.ac.il (N.B.); yaelho@ekmd.huji.ac.il (Y.H.-H.); 2The Institute of Dental Sciences, Faculty of Dental Medicine, Hebrew University of Jerusalem, Jerusalem 9112001, Israel

**Keywords:** *S. mutans*, trans-cinnamaldehyde, antibacterial, antibiofilm, preformed biofilms

## Abstract

*Streptococcus mutans* (*S. mutans*) is the main cariogenic bacterium with acidophilic properties, in part due to its acid-producing and -resistant properties. As a result of this activity, hard tooth structures may demineralize and form caries. Trans-cinnamaldehyde (TC) is a phytochemical from the cinnamon plant that has established antibacterial properties for Gram-positive and -negative bacteria. This research sought to assess the antibacterial and antibiofilm effects of trans-cinnamaldehyde on *S. mutans*. TC was diluted to a concentration range of 156.25–5000 μg/mL in dimethyl sulfoxide (DMSO) 0.03–1%, an organic solvent. Antibacterial activity was monitored by testing the range of TC concentrations on 24 h planktonic growth compared with untreated *S. mutans*. The subminimal bactericidal concentrations (MBCs) were used to evaluate the bacterial distribution and morphology in the biofilms. Our in vitro data established a TC MBC of 2500 μg/mL against planktonic *S. mutans* using a microplate spectrophotometer. Furthermore, the DMSO-only controls showed no antibacterial effect against planktonic *S. mutans*. Next, the sub-MBC doses exhibited antibiofilm action at TC doses of ≥625 μg/mL on hydroxyapatite discs, as demonstrated through biofilm analysis using spinning-disk confocal microscopy (SDCM) and high-resolution scanning electron microscopy (HR-SEM). Our findings show that TC possesses potent antibacterial and antibiofilm properties against *S. mutans*. Our data insinuate that the most effective sub-MBC of TC to bestow these activities is 625 μg/mL.

## 1. Introduction

Dental caries is the most prevalent infectious disease of the oral cavity. Biofilms formed by microorganisms over teeth and gum surfaces play an important causative role.

It is a well-known fact that biofilm prevents antibiotics from inhibiting bacterial growth and, thus, may contribute to the development of antibiotic resistance. Moreover, the widespread use of antibiotics results in an increase in more virulent and resistant bacteria. The emergence of antimicrobial agents’ resistant pathogenic bacteria has become a severe health concern [1]; hence, new antibacterial agents are always beneficial.

One of the main etiological factors of dental caries is *Streptococcus mutans (S. mutans)* [2,3,4,5,6], a Gram-positive facultative anaerobic bacterium [7,8].

The pathogenicity of *S. mutans* is inseparably associated with its ability to form biofilms on solid surfaces, such as implants and catheters [2].

*S. mutans* has the ability to form extracellular polysaccharides (EPS), essential components in dental biofilms [9]. *S. mutans* also generates glucan-binding proteins, which are assumed to encourage adherence to matrix glucans and influence the biofilm’s general structure [10,11]. Moreover, it has the ability to thrive in an acidic environment (aciduric) and metabolize carbohydrates, which produce lactic acids, leading to the development of dental tissue demineralization (acidogenic) [2,7].

Natural materials, herbs, spices, essential oils, etc., have been used in traditional medicine for many years [12]. The cinnamon plant is one of these natural components. The main constituents of cinnamon essential oils and extracts are cinnamaldehyde, eugenol, phenol, and linalool [13]. Trans-cinnamaldehyde (TC) is a phytochemical from the cinnamon plant.

TC, an α- and β-unsaturated aromatic aldehyde, offers numerous advantages. It can be obtained from a variety of sources, such as various vegetables and fruits, and it has a low cost and low cytotoxicity [14].

Many studies have shown that cinnamaldehyde has medical and agricultural benefits. Gomez et al. [15] proved that it inhibited fungal growth and aflatoxin production in food. Kim et al. [16] showed how TC induces anti-inflammatory effects on the activation of macrophages stimulated via LPS. Lu et al. [17] summarized cinnamaldehyde’s therapeutic effects and its related cardiovascular protective mechanisms.

In nature, cinnamaldehyde occurs as a trans-stereoisomer, namely, (2E)-3-phenylprop-2-enal or trans-cinnamaldehyde (TC) [18]. However, when TC is exposed to air, it is quite unstable because the reactive unsaturated aldehyde is oxidized to cinnamic acid [19]. This is due to its chemical structure of unstable bonds and being a member of the aldehyde group [20].

This unsaturated aldehyde is the component of cinnamon that has established antibacterial properties for Gram-positive and -negative bacteria [21]. TC has been shown to effectively inhibit many microorganisms’ growth and has been reported to inhibit toxin production by micro-organisms [21].

Mu et al. recently reported their attempts to use chitosan-based nanocapsules loaded with TC against an *S. mutans* biofilm [22]. The nanoparticles adsorb to the bacterial membrane and act against *S. mutans*.

Utilizing plants for antimicrobial agents is a reasonable strategy due to their natural production of a wide range of secondary metabolites that provide defenses against microbial invasions of the host. Additionally, unique chemical entities that exist in plants, such as complex structures, can limit the attempt to artificially produce them.

This capacity could enable these natural antibacterials to activate different modes of action in comparison to conventional antibacterial agents, thereby helping to reduce microbial resistance [23].

In the present study, our general objective was to assess and establish the antibacterial effect of trans-cinnamaldehyde on *S. mutans* and its in vitro antibiofilm activity. We hypothesized that the TC tested would possess a potent antibacterial effect against *S. mutans*.

## 2. Materials and Methods

Trans-cinnamaldehyde (TC) (C_9_H_8_0; C_6_H_5_CH=CHCHO), also called trans-3-Phenyl-2-propenal (trans-cinnamaldehyde 99%, Sigma-Aldrich, Jerusalem, Israel), was purchased. TC’s molecular structure is shown in Figure 1 (molecular weight, 132.16 g/mol). The TC was diluted progressively to the following concentrations: 156.25 μg/mL, 312.5 μg/mL, 625 μg/mL, 1250 μg/mL, 2500 μg/mL, and 5000 μg/mL in dimethyl sulfoxide (DMSO) 0.03–1% (an organic solvent).

### 2.1. Bacterial Growth and Culture Conditions

Planktonic *S. mutans* UA159 was grown overnight at 37 °C in brain–heart infusion broth (BHI, Acumedia, Lansing, MI, USA) in 95% air/5% CO_2_. An OD600 nm of 0.1 was determined as the initial concentration. The bacterial cultures were treated with the concentrations of TC mentioned above. Untreated bacteria served as the control, while chlorhexidine served as a positive control and BHI alone as a clear control. For kinetic studies, samples of *S. mutans* were treated with increasing concentrations of TC (156.25–5000 μg/mL) and the OD650 nm was measured every 15 min for 24 h using a microplate reader (Tecan M200, Tecan Trading AG, Männedorf, Switzerland) at 37 °C.

### 2.2. Viable Count Evaluations

The colony-forming unit (CFU) assay was carried out following various incubation times (0, 4 h, 8 h, 12 h, 16 h, 20 h, and 24 h) with TC in numerous concentrations (156.25–5000 μg/mL). Progressive serial dilutions were conducted on both untreated and treated samples. This process entailed transferring 100 μL from one sample to another tube containing 900 μL of phosphate-buffered saline (PBS). Following thorough vortexing, 10 μL of the bacterial suspension was subsequently plated in triplicate on brain–heart infusion (BHI) agar plates. The plates were then incubated overnight at 37 °C under conditions of 5% CO_2_.

### 2.3. Agar Diffusion Test (ADT)

The agar diffusion test is also known as the agar contact method. Following the inoculation of 100 μL 0.1 OD650 nm *S. mutans* on BHI–agar plates, we placed 10 μL of our different TC concentrations (156.25–5000 μg/mL) onto sterilized circular chromatography papers. This method involved the transfer, through diffusion, of TC to the plates inoculated with *S. mutans* and placed in a closed jar with a CO_2_ bag in the incubator at 37 °C. After 24 h, we evaluated the growth inhibition halo around the circular papers.

### 2.4. Biofilm Biomass Evaluation Using Crystal Violet (CV) Staining

For biofilm growth, *S. mutans* UA159 was grown overnight at 37 °C in 95% air/5% CO_2_ in brain–heart infusion and diluted to an OD600 nm of 0.1 in BHI containing 2.5% sucrose (BHIS), and then treated with increasing concentrations of TC (156.25–5000 μg/mL) with a positive control (chlorhexidine) and an untreated control for 24 h.

The resultant biofilms underwent staining with 200 μL of 0.1% crystal violet (CV), prepared from a 0.4% Gram’s crystal violet solution (Merck, EMD Millipore Corporation, Billerica, MA, USA) and diluted with double-distilled water (DDW). Following a 15 min incubation at room temperature, the CV solution was aspirated, and the wells were subjected to two washes with DDW and subsequently air-dried overnight. The extraction of the CV stain ensued through the addition of 150 μL of 33% acetic acid to the wells, accompanied by a 20 min incubation on an orbital shaker. The quantification of biofilm biomass was determined by measuring the absorbance at 595 nm using the M200 Tecan plate reader (Tecan Trading AG, Männedorf, Switzerland).

### 2.5. Biofilm Biomass Evaluation via Tetrazolium Reduction Assay (MTT Metabolic Assay)

The MTT assay is a colorimetric assay measuring bacterial metabolic activity. Following the *S. mutans* UA159 biofilm growth and incubation in the presence or absence of TC as described above, the MTT metabolic assay was performed. A 0.5 mg/mL MTT solution (Calbiochem, Darmstadt, Germany) in phosphate-buffered saline (PBS) was added to the biofilms in 96-well plates at a volume of 50 μL. After a 1 h incubation period at 37 °C, the wells underwent PBS washing. After an additional 1 h incubation period, absorbance readings were taken at 570 nm using the M200 Tecan plate reader (Tecan Trading AG, Männedorf, Switzerland) for quantitative assessment.

### 2.6. Metabolic Activity Assay (MTT) with Preformed Biofilms

*S. mutans* biofilm was grown in the manner previously mentioned, and then incubated for 4 h at 37 °C before being treated with increasing concentrations of TC (156.25–5000 μg/mL) with a positive control (chlorhexidine) and an untreated control for 24 h.

Then, the biofilms were washed twice with PBS, and MTT assays were performed. The wells were washed with PBS after 1 h of incubation at 37 °C. After another 1 h of incubation, the absorbance was measured at 570 nm using the M200 Tecan plate reader (Tecan Trading AG, Männedorf, Switzerland).

### 2.7. Biofilm Analysis via Spinning-Disk Confocal Microscopy (SDCM)

Spinning-disk confocal microscopy (SDCM) was employed to examine the biofilm architecture post-treatment with the assessed material, discerning live/dead bacterial presence and extracellular polysaccharides (EPS). The biofilms, cultivated on hydroxyapatite discs in 24-well tissue culture plates for 24 h in the absence or existence of varying TC concentrations, underwent double PBS washes.

Subsequently, staining ensued with 3.3 μM SYTO 9, 10 μg/mL propidium iodide (PI), and 10 μg/mL Alexafluor647-conjugated dextran 10,000 for 20 min at room temperature.

SYTO 9’s green fluorescence (488 nm excitation, 515 nm emission) visualized both live and dead bacteria, while PI’s red fluorescence (543 nm excitation, 570 nm emission) specifically detected dead bacteria. Dextran’s blue fluorescence highlighted extracellular polysaccharides (EPS), presenting a fluorescence wavelength of 640 nm excitation and 665 nm emission for Alexafluor647. This resulted in a dual fluorescence, with live bacteria emitting green light and dead bacteria emitting both green and red light.

Utilizing the Nikon Yokogawa W1 spinning-disk confocal microscope, the samples were observed for thickness and bacterial vitality, capturing optical cross-sections at 2.5 μm intervals from the biofilm’s bottom to its apex.

### 2.8. High-Resolution Scanning Electron Microscopy (HR-SEM)

The biofilms were cultured on sterile hydroxyapatite discs, both with and without varying concentrations of TC. Following 24 h of incubation, the hydroxyapatite discs underwent rinsing with double-distilled water (DDW) and were subsequently fixed in 4% glutaraldehyde in DDW for 40 min. After an additional DDW wash, the discs were air-dried at room temperature. Subsequently, the hydroxyapatite discs were affixed to a metal stub, subjected to iridium sputter coating, and examined through a Magellan XHR 400 L high-resolution scanning electron microscope (Magellan XHR 400 L, FEI Company, Hillsboro, OR, USA). Three specimens from each treatment group were prepared and analyzed under SEM to assess the impact of TC on biofilm formation.

### 2.9. Statistical Analysis

Absorbance measurements were graphically represented, yielding bacterial growth curves for each well. The statistical examination concentrated on the linear segment of the logarithmic growth phase, where the slope denoted the bacterial growth rate and the intercept reflected the total viable bacterial count. The data analysis employed one-way ANOVA followed by Tukey’s post hoc test. The level of significance was *p* < 0.05.

## 3. Results

### 3.1. TC Inhibits Planktonic Growth of S. mutans

An antibacterial action on planktonic *S. mutans* was observed for all test groups, as can be seen in Figure 2. Using 156.25 μg/mL, a partial antibacterial effect was observed. A statistically significant difference was found between 156.25 μg/mL and all other test groups with higher TC concentrations. No significant difference was found between the higher concentrations. To eliminate the bias of DMSO being the reason for the antibacterial activity, we repeated the same experiment and used DMSO as the antibacterial agent and we observed no antibacterial activity.

### 3.2. TC Inhibited S. mutans Colony Formation

The total bacterial growth inhibition was observed using 5000 μg/mL after 4 h and using 2500 μg/mL concentrations after 12 h. Hence, the MBC dose observed was 2500 μg/mL. Furthermore, the 312.5–2500 μg/mL concentrations reduced colony formation for 24 h with respect to the untreated *S. mutans* control, as can be seen in Figure 3. The antibacterial effect was time- and dose-dependent.

### 3.3. TC Demonstrated No Visible Halo with the Agar Diffusion Test (ADT)

No visible inhibition halo around the chromatography papers was observed with any of our TC concentrations. Representative photos of plates inoculated with *S. mutans* and treated using the chlorhexidine control, 312.5 μg/mL, and 5000 μg/mL are shown in Figure 4.

### 3.4. TC Exhibited Antibiofilm Effects

The TC concentrations ≥ 625 μg/mL exhibited a statistically significant antibiofilm effect compared to the untreated *S. mutans* control with both the CV and MTT assays, as can be seen in Figure 5 and Figure 6, respectively. It can be seen that concentrations of 312.5 μg/mL and 125.25 μg/mL had no significant effect on biofilm biomass compared to the untreated *S. mutans*.

### 3.5. TC Exhibited Antibiofilm Activity with Preformed Biofilms

Following the initial formation of the biofilm, the treatment with TC concentrations of ≥312.5 μg/mL significantly reduced the biofilm formation compared to the untreated *S. mutans* control, as can be seen in Figure 7.

### 3.6. Live/Dead Bacteria and EPS Modification by TC

Microscopic cross-sectioned images of the biofilms on HA discs are shown in Figure 8, Figure 9, Figure 10 and Figure 11.

EPS and SYTO9 staining showed a reduction in live bacteria when the biofilms were treated with the sub-MBC doses that were previously observed.

Furthermore, the PI staining demonstrated that the TC caused an elevated dead bacteria population compared to the control.

### 3.7. TC Altered Bacterial Morphology and Density

In the control, bacteria with intact membranes and dividing cells can be observed. The sub-MBC TC doses reduced the *S. mutans* density compared to the untreated control, as can be seen in Figure 12a–h. The bacteria in the different treatment groups showed changes in morphology with abnormal cell division, as highlighted in Figure 12. The use of increasing concentrations of the tested TC caused morphological changes and reduced the numbers of bacteria.

## 4. Discussion

Trans-cinnamaldehyde is an effective and fast-acting antibacterial agent that can cause total bacterial growth inhibition using high concentrations. Partial bacterial growth inhibition was observed using all of the tested concentrations.

*S. mutans* is a Gram-positive bacterium that is a key contributor to dental caries and the formation of dental biofilms. It plays an important role in the production of pathogenic dental biofilms owing to its ability to produce extracellular polysaccharides [9]. Gram-positive bacteria are more susceptible to phytochemicals and their constituents than Gram-negative bacteria due to the less complex cell walls of Gram-positive bacteria [24]. Moreover, Gram-positive bacteria allow hydrophobic molecules, including TC, to easily pass through the cell membrane and operate on both the cell wall and the cytoplasm [19].

To evaluate the antibacterial and antibiofilm activity of TC on *S. mutans*, we used our obtained sub-MBC doses. This is because the use of higher TC concentrations will lead to a bacteriocidic effect, which may cause dramatic changes in natural flora, leaving no bacteria to evaluate.

Our MBC against planktonic *S. mutans* was 2500 μg/mL, which was higher than the 2000 μg/mL value obtained by He et al. [14]. This could be due to technical differences such as the OD and TC concentrations used.

In terms of the antibiofilm activity, the 625 μg/mL TC dose was the lowest effective concentration, and this dose can be used for future translational studies.

We also observed a significant antibiofilm activity with the preformed (treatment) biofilm using our sub-MBC doses ≥ 312.5 μg/mL, similar to the results obtained in the metabolic assays. The ability of the TC to affect the biofilm after it was established was proven. These findings set the basis for future in vivo studies.

Additionally, biofilm formation is associated with quorum-sensing activity [25]; hence, it may be reasonable to assume that this biofilm inhibition may lead to the downregulation of quorum-sensing systems (an intercellular communication system), making *S. mutans* more prone to antibacterial agents and less resistant.

We observed no visible inhibition halo around disks with our TC concentrations; this could be due to the low solubility and hydrophobic nature of TC, which inhibits diffusion on sterilized circular chromatography papers.

Hydroxyapatite, a calcium phosphate compound with a calcium-to-phosphorus ratio of 1:67 and the chemical formula Ca_10_ (PO_4_) OH_2_, is the main inorganic component of teeth and bone [26]. The microscopic assays HR SEM and SDCM were conducted on hydroxyapatite disks, an essential component of tooth enamel, in order to simulate the effect in the oral cavity as much as possible. Through HR SEM and SDCM, morphological cell wall changes were observed, as well as bacterial aggregation and syncytium-like cell-wall fusion. No visible cell division was observed.

The use of 156.25 μg/mL and 312.5 μg/mL concentrations resulted in the partial inhibition of *S. mutans* growth, as can be seen in Figure 2, Figure 3 and Figure 7. Hence, the dramatic increase in the number of dead cells stained with PI was due to an overall higher number of dead cells. When using concentrations of 625 μg/mL and 1250 μg/mL, as expected, we observed more cell death and some (albeit less) biofilm activity. The 2500 μg/mL and 5000 μg/mL concentrations caused total bacterial kill after 24 h, and, thus, were not included in the following imaging experiments due to the fact that these concentrations resulted in almost no live bacteria, causing 99.9% of cell death at ≥ the MBC dose. When thinking of human clinical applications, we need to use the lowest TC concentration possible.

In Figure 10f, an increase in EPS staining was observed, which was higher compared to the lower TC concentrations. This higher fluorescence observed in Figure 10f is a three-dimensional representation of just one well, and the relative fluorescent intensities of the area under the curve are a better representation of our quantitative EPS measurement for the 1250 μg/mL concentration (Figure 11C), which showed reduced EPS formation for all treatment groups, including 1250 μg/mL, compared to our untreated control.

EPS constitutes a diverse matrix of polymers, encompassing polysaccharides, proteins, glycoproteins, nucleic acids, phospholipids, and humic acids. Its documented role involves the establishment of a gel-like network, promoting bacterial cohesion within biofilms, facilitating biofilm adhesion to surfaces, and providing protection against adverse environmental conditions. Despite these recognized functions, the precise factors governing the augmentation of EPS formation remain incompletely elucidated. Polysaccharides can also provide protection from a wide range of stresses and it may be suggested that the stress formed by the TC can cause a temporary increase in EPS.

Balasubramanian et al. [27] suggested a synergistic effect against cariogenic bacteria when combining TC with chlorohexidine or fluoride.

As reported in the literature, cinnamon essential oils may also be beneficial in the treatment of halitosis, while reducing biofilm formation [13]. The use of the sub-MBC that was evaluated in this research may also be useful for the treatment of halitosis and caries, with a possible minimal effect on the microbial oral population. This assumption should be further investigated clinically.

Previous study found that TC decreased *S. mutans* activity and affected biofilm formation. TC increased the cell surface hydrophobicity and reduced bacterial aggregation, thus affecting the process of biofilm formation. It also inhibited the acid production and acid tolerance of *S. mutans*. In the presence of cinnamaldehyde, the gene expression in the biofilms was reduced [14].

Hence, TC, derived from a natural substance, has shown significant potential to serve as an antimicrobial modality, particularly in inhibiting caries-promoting bacterial growth.

Dentifrices comprise several antimicrobial agents, such as fluoride and zinc salts. Given the antibacterial properties we have shown, it may be beneficial to add TC to dentifrice, lozenges, mouthwashes, etc., as a natural antimicrobial agent.

An important issue that must be considered is the large amount of evidence in the literature regarding allergy to cinnamon [28,29,30]. Cinnamaldehyde is a known sensitizer in humans and it has been reported that cinnamaldehyde is a frequent cause of allergic reactions. In human dermatological studies, the level for cinnamaldehyde sensitization has been set at 0.5% [21]. Thus, this highlights the need for research into cinnamaldehyde derivates using the lowest concentration possible to achieve good antibacterial effects. Therefore, care must be taken when using TC as a caries-preventive modality.

Further studies are needed to prove the anti-cariogenic potential of TC, with or without different antibacterial therapeutics.

## 5. Conclusions

In conclusion, we observed antibacterial and antibiofilm effects using sub-MBC TC doses on *S. mutans.*

The antibacterial effect was observed at all concentrations. However, the 625 μg/mL concentration seemed to be the lowest concentration exhibiting antibacterial and antibiofilm activities and should be furthered tested for future preventive and treatment modalities.

The application of these TC activities can be numerous, including as a potential novel anti-cariogenic material.

## Figures and Tables

**Figure 1 pharmaceutics-16-00113-f001:**
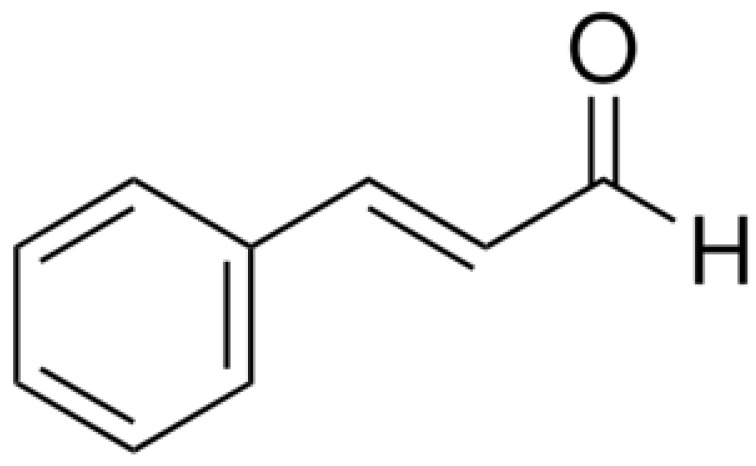
The molecular structure of trans-cinnamaldehyde (taken from the Sigma-Aldrich^®^ website). Linear formula, C6H5CH=CHCHO; molecular weight, 132.16 g/mol.

**Figure 2 pharmaceutics-16-00113-f002:**
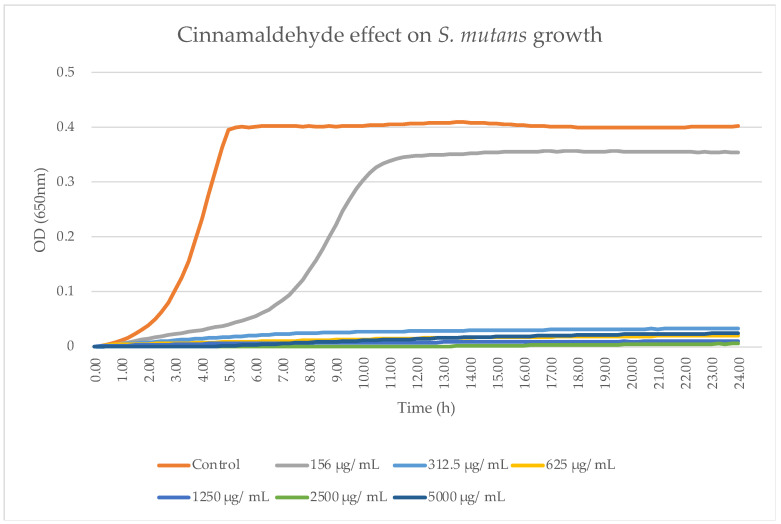
Antibacterial activity of TC. *S. mutans* kinetic growth was analyzed after 24 h in a microplate reader. Each point on the curve represents the mean absorbance (OD650 nm) concurrently measured in four wells that were uniformly prepared within a microtiter plate. Bacterial growth was not found at concentrations of 625 μg/mL and above (each experiment was repeated twice).

**Figure 3 pharmaceutics-16-00113-f003:**
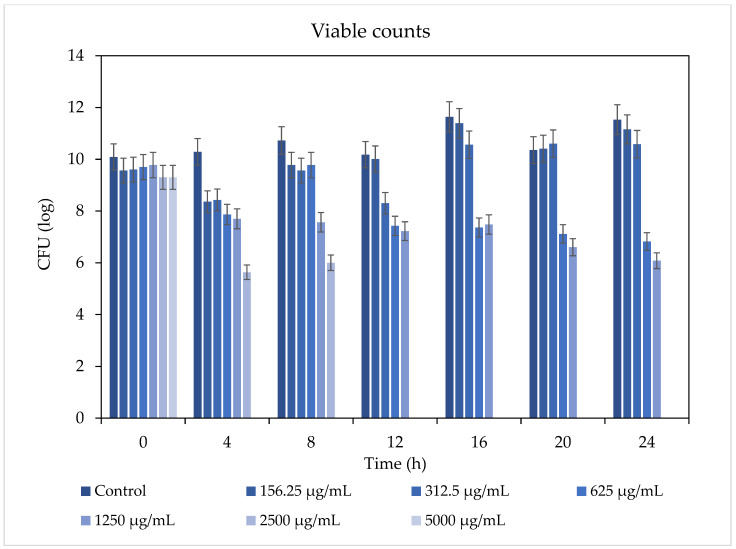
Viable counts after direct contact at different incubation times. The antibacterial activity in planktonic culture with *S. mutans* showed total growth inhibition using ≥2500 μg/mL. Using 5000 μg/mL caused total growth inhibition after only 4 h. The antibacterial effect was time- and dose-dependent.

**Figure 4 pharmaceutics-16-00113-f004:**
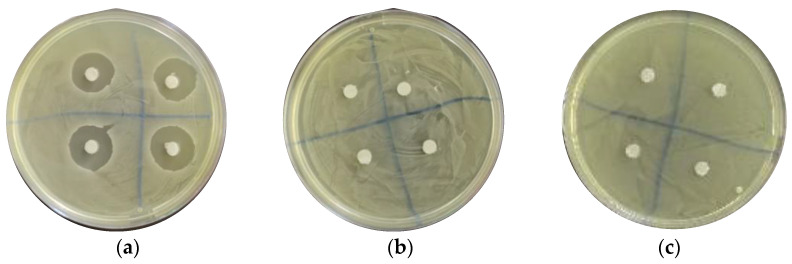
TC demonstrated no visible halo with the agar diffusion test. (**a**) Chlorhexidine 1.25% control shows significant halo. TC shows no bacterial inhibition for both 312.5 μg/mL (**b**) and (**c**) 5000 μg/mL.

**Figure 5 pharmaceutics-16-00113-f005:**
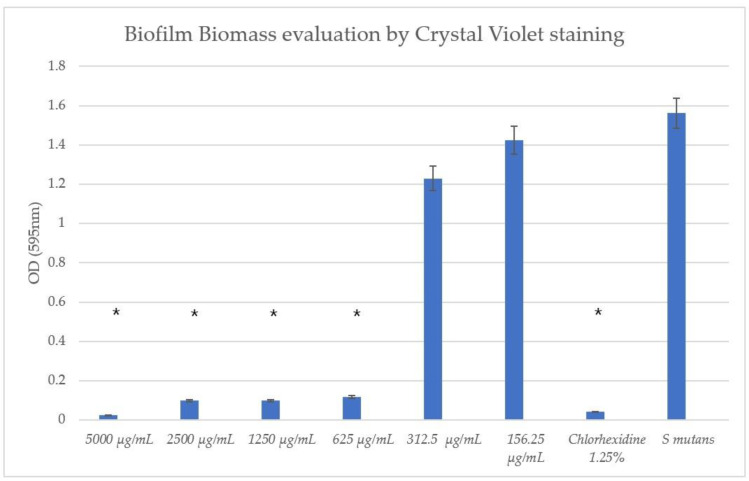
Biofilm colorimetric assay with CV. The crystal violet biofilm mass staining showed statistically significant effects using ≥625 μg/mL concentrations of TC. * *p* < 0.05 (eight wells per concentration, three repetitions).

**Figure 6 pharmaceutics-16-00113-f006:**
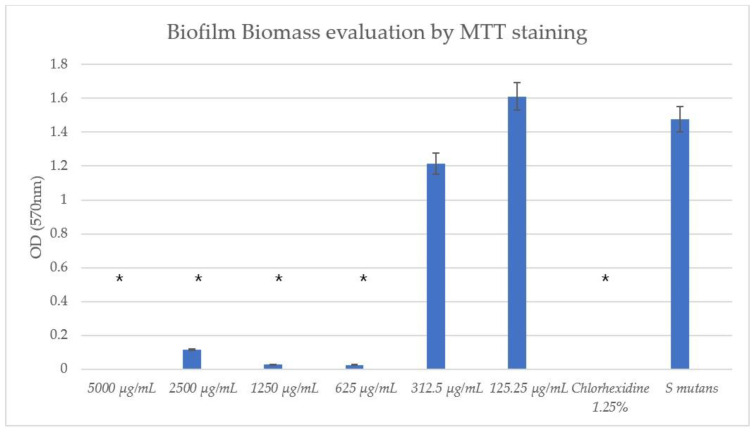
Biofilm colorimetric assay with MTT. The MTT assay showed statistically significant effects after 1 h of incubation using ≥625 μg/mL concentrations of TC. * *p* < 0.05 (eight wells per concentration, three repetitions).

**Figure 7 pharmaceutics-16-00113-f007:**
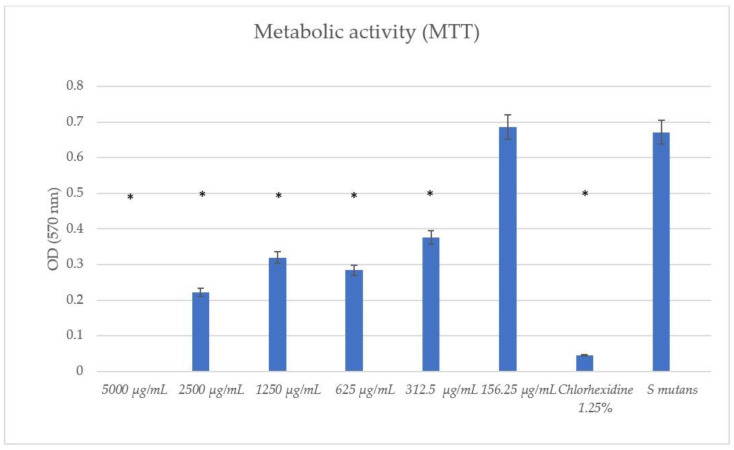
Metabolic activity assay (MTT) with preformed biofilms. This assay showed statistically significant antibiofilm effects using ≥312.5 μg/mL concentrations of TC. * *p* < 0.05 (eight wells per concentration, three repetitions).

**Figure 8 pharmaceutics-16-00113-f008:**
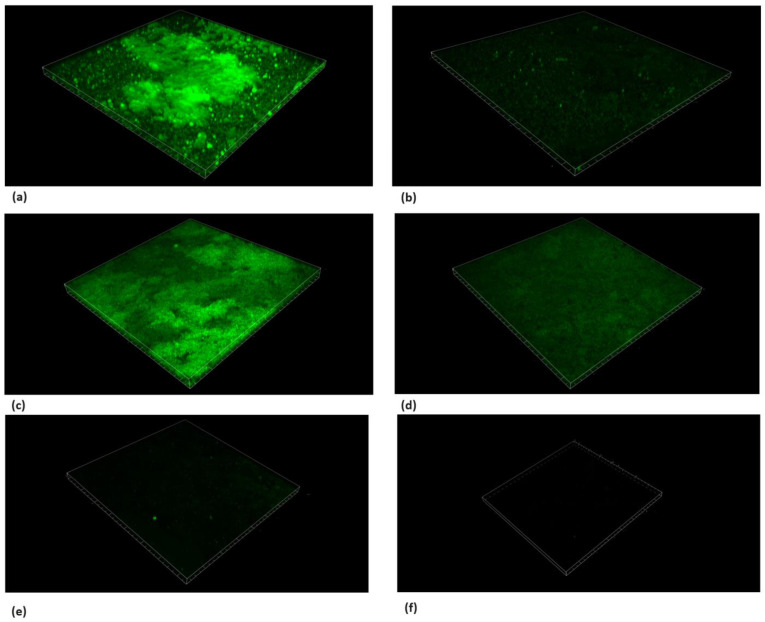
Live/dead staining with SYTO9. Microscope’s three-dimensional images of biofilm formed on hydroxyapatite discs. Surface dimensions are 1493.91 μm × 1493.91 μm. (**a**) *S. mutans*, (**b**) chlorhexidine 1.25%, (**c**) TC 156.25 μg/mL, (**d**) TC 312.5 μg/mL, (**e**) TC 625 μg/mL, and (**f**) TC 1250 μg/mL. Antibiofilm effect was observed using concentrations ≥625 μg/mL.

**Figure 9 pharmaceutics-16-00113-f009:**
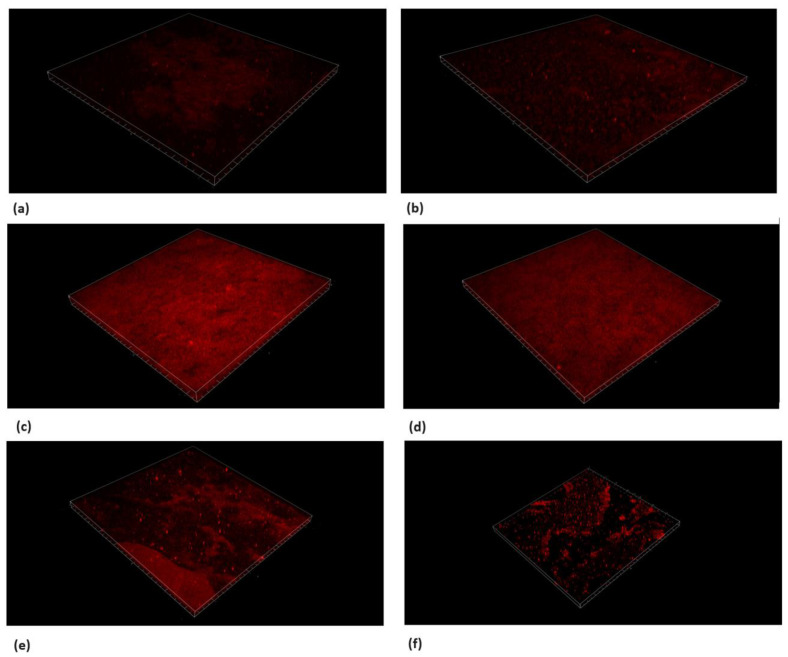
Live/dead staining with PI. Microscope’s three-dimensional images of biofilm formed on hydroxyapatite disks. Surface dimensions are 1493.91 μm × 1493.91 μm. Images show the dead bacteria population using the sub-MBC of TC. An antibiofilm effect was observed using concentrations ≥ 625 μg/mL. (**a**) *S. mutans*, (**b**) chlorhexidine 1.25%, (**c**) TC 156.25 μg/mL, (**d**) TC 312.5 μg/mL, (**e**) TC 625 μg/mL, and (**f**) TC 1250 μg/mL.

**Figure 10 pharmaceutics-16-00113-f010:**
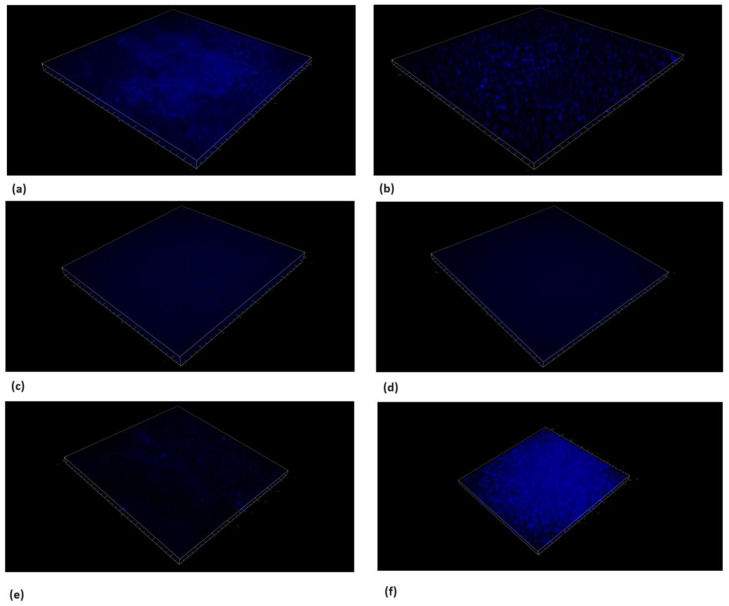
Extracellular polysaccharide (EPS) staining. Microscope’s three-dimensional images of biofilm formed on hydroxyapatite discs. Surface dimensions are 1493.91 μm × 1493.91 μm. Dextran blue fluorescence dye was used to demonstrate the effect of TC on the biofilm. The use of TC caused a reduction in the blue fluorescence compared to the untreated control. (**a**) *S. mutans*, (**b**) chlorhexidine 1.25%, (**c**) TC 156.25 μg/mL, (**d**) TC 312.5 μg/mL, (**e**) TC 625 μg/mL, and (**f**) TC 1250 μg/mL.

**Figure 11 pharmaceutics-16-00113-f011:**
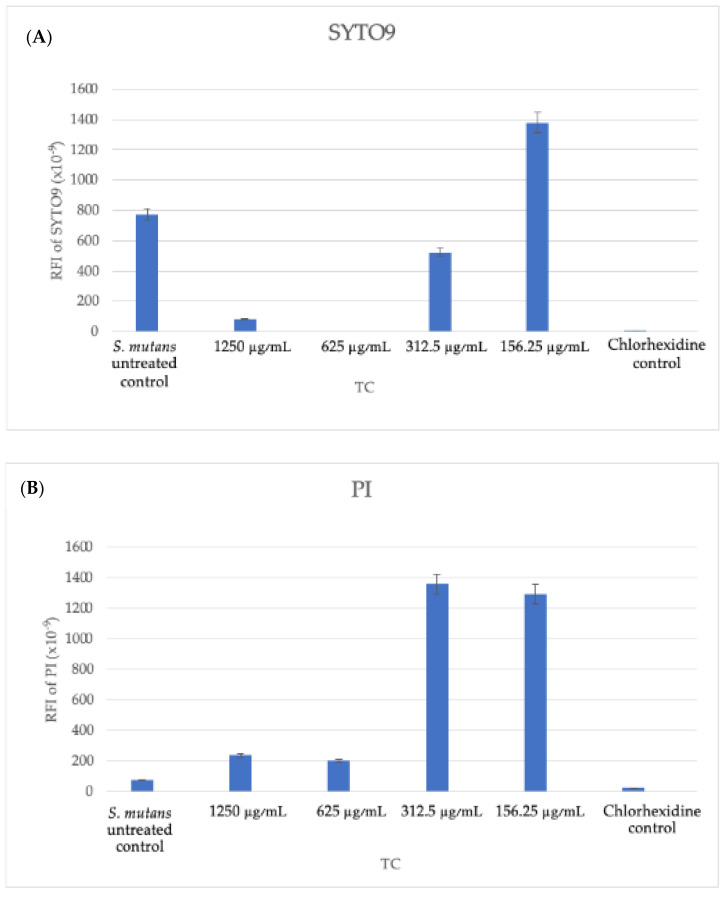
Spinning-disk confocal microscopy (SDCM) analysis. Quantitative analysis of the staining with SYTO9 (all bacteria), PI (dead bacteria), and dextran (extracellular polysaccharides) using relative fluorescence intensities (RFIs). Quantifications of the areas under the curves of SYTO 9 (**A**), PI (**B**), and dextran (**C**) in every biofilm layer, recorded every 2.5 μm. The relative fluorescence intensities (RFIs) of PI/SYTO 9/dextran in the biofilms of untreated control (**D**) and 625 μg/mL TC-treated samples (**E**).

**Figure 12 pharmaceutics-16-00113-f012:**
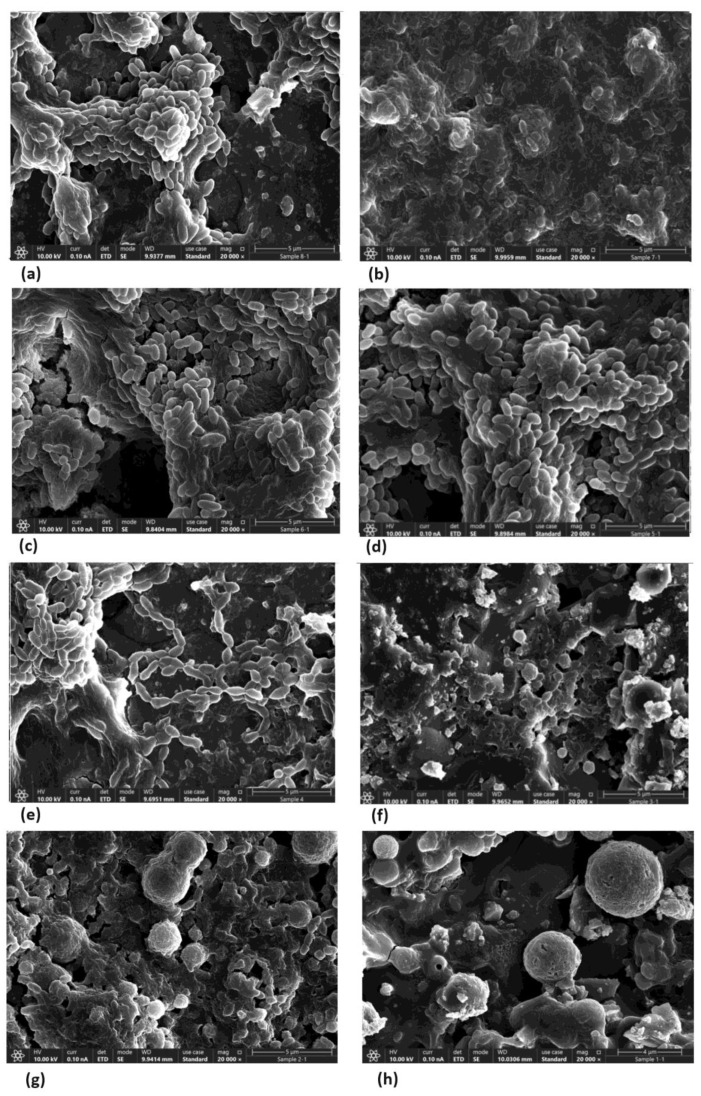
SEM images of *S. mutans* on hydroxyapatite discs after 24 h of incubation. (**a**) *S. mutans* control. Dividing bacterial cells with intact membranes and biofilm formation can be observed. (**b**) Chlorhexidine control. Cell wall disruption and lysis. (**c**,**d**) Images at 156.25 and 325 μg/mL, respectively. Morphologic changes in bacterial cell wall can be observed. (**e**) 625 μg/mL. Further morphologic changes in the bacterial cell wall and bacterial aggregation. (**f**–**h**) Images at 1250, 2500, and 5000 μg/mL, respectively. Most of the bacteria show changes in morphology with no visible signs of cell division. Further morphological changes include cell wall disruption and lysis. Specimens were observed using a high-resolution scanning electron microscope (Magellan XHR 400 L, FEI Company, Hillsboro, OR, USA, The Hebrew University of Jerusalem, Jerusalem, Israel).

## Data Availability

Data is contained within the article, and on request from the authors.

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
