# Peer review of "Trans-Cinnamaldehyde—Fighting Streptococcus mutans Using Nature"

_pharmaceutics, 2024, doi:10.3390/pharmaceutics16010113_

Round 1

Reviewer 1 Report

Comments and Suggestions for Authors

This study evaluated the effects of trans-cinnamaldehyde in different concentration inhibiting planktic S. mutans and its biofilm formation. Obviously, trans-cinnamaldehyde of 625 μg/mL showed  antibacterial and anti-biofilm activities. The major limitation of this study could be the experimental design is relatively simple, considering that several other literatures have reported the anti-S.mutans effect of cinnamaldehyde. Furthermore, some points should be revised, as follows:

1.     Please supplement information about the chemical structure of trans-cinnamaldehyde in figures.

2.     Please supplement the results of the agar diffusion test in figures.

3.     The presentation of results in figures should be clear and easy to read. Please revise the layout of the figures.

4.     Why choose 1.25% concentration of chlorhexidine as a positive control, when the commonly used concentrations are 0.2% or 0.12%.

5.     What about the effect of trans-cinnamaldehyde on eliminating mature biofilms? Has the author conducted relevant evaluations?

6.     In SDCM, what was the excitation and emission wavelength of Alexafluor647?  In addition, was there a large overlap between the wavelength range of Alexafluor647 and PI in co-staining?

7.     In Live/Dead SDCM, SYTO9 can stain both dead and live bacteria while PI only stains dead bacteria. However, compared with PI staining, the REI value of SYTO9 is much lower in Figure 9b. How to explain this phenomenon?

8.     Likewise, the fluorescence intensity 1250 μg/mL trans-cinnamaldehyde is much higher than that at low concentrations for EPS staining. Unfortunately, this study did not discuss this result.

9.     In discussion, the authors mentioned that trans-cinnamaldehyde is unstable and susceptible to oxidation. So, does this mean that its use as an anticaries agent is easy to be invalid?

10.  Minor comments: The writing requires proofing.

    Line 10,13 and 17 in AbstractThere should be a space between the number and measure. Please also revise elsewhere in the paper.

    Statistical P values should be written in italics in full paper.

Comments on the Quality of English Language

Moderate editing of English language required

Reviewer 2 Report

Comments and Suggestions for Authors

In this paper, author  investigates the antibacterial and antibiofilm characteristics of Trans-Cinnamaldehyde (TC) against Streptococcus mutans (S. mutans), which is known for its acid-producing and acid-resistant properties and is implicated in the production of dental caries Trans-Cinnamaldehyde is a naturally occurring chemical found largely in cinnamon oil that has antibacterial effects. Streptococcus mutans is a bacteria that is a major cause of tooth decay.  But still title need some modification looking incomplete in end with term of nature 

Abstract:  This section is succinct and clear, providing a simple comprehension of the study's aims, methodology, and results. But the incorporation of controls and diverse assessment methodologies, such as SDCM and HR-SEM, adds credibility to the findings. A more complete comprehension of the process, on the other hand, would be advantageous for a thorough study.  The article demonstrates the potential of TC in combating S. mutans, laying the groundwork for further research or prospective usage in dental care products. Because the study was conducted in vitro, extrapolating these findings to in vivo circumstances should be done with caution. The provided  information on replication, statistical analysis, and external validity would be useful in completely appraising the report's conclusions.

general query 

What are the main goals of the current study's evaluation of the antibacterial and anti-biofilm activities of TC?

What is the study's major premise, and how does it connect to the well-known characteristics of TC?

What scientific strategy is most likely to be used, given the objectives and hypothesis, to evaluate the antibacterial effects of TC on S. mutans?

How would the study likely handle the difficulties provided by biofilm development in assessing the effectiveness of TC?

How does S. mutans contribute most to the emergence of dental caries?

What features of S. mutans make it a substantial contributor to the development of biofilm and, consequently, tooth decay?

Why do biofilms present a serious obstacle to using traditional antibiotic therapies?

What role do the form and makeup of biofilms play in the emergence of antibiotic resistance?

What are Trans-Cinnamaldehyde's (TC) known antibacterial properties?

How can TC, a substance produced from the cinnamon plant, fit the bill as a possible antibacterial therapy option?

Based on past investigations, how does TC interact with bacterial membranes, specifically those of S. mutans?

What ecologically sound approach does utilising plants as sources of antibacterial agents represent?

What benefits can secondary metabolites originating from plants offer in terms of defending against microbial invasion?

Results and discussion okay 

Image: very good

Oveal manuscript is good  

Comments on the Quality of English Language

not reqired

Reviewer 3 Report

Comments and Suggestions for Authors

This paper studied the anti-bacterial and anti-biofilm effect of trans-cinnamaldehyde on S. mutans. However, some problems must be solved before it is considered for publication.

1.Line 348, The author mentioned that Zhiyan et al. (14) also determined that the MBC of TC on S. mutans was 2000µg/ml, which means that Zhiyan et al. also did the related work of TC on S. mutans. However, the reference of Zhiyan et al (14) does not appear in the References, which is a wrong citation. The author must provide this reference, and explain how many similarities and differences exist between the work in this reference and the work in this study.

2. The format of references is extremely confusing and incorrect, which should be rearranged and checked.

3. The formats of all drawings (Figures 12345 and 9) should be unified and beautified.

4. The time units in Figure 1 and Figure 2 should be consistent, hrs or h?

5. In the legend of Figure 1, 156 g/ml should be 156.25 µg/ml.

6. In the discussion, the content of line 319-line 330 should appear in the Introduction.

6. In the discussion, the author should give more discussion about the antibacterial mechanism of trans-cinnamaldehyde on S. mutans.

Reviewer 4 Report

Comments and Suggestions for Authors

In this study, the authors investigated the antibacterial and anti-biofilm effects of trans-cinnamaldehyde (TC) on S. mutans. They determined that the minimum bactericidal concentration (MBC) of TC against planktonic S. mutans was 2500 μg/mL, and a sub-MBC concentration of 625 μg/mL was found to be the lowest effective concentration for anti-biofilm effects. However, several aspects of this study lack novelty, and concerns about the overall manuscript quality arise. Here are my comments:

1.      The antibacterial and anti-biofilm effects of TC, including MBC, have previously been documented in the literature (Front Microbiol. 2019, doi: 10.3389/fmicb.2019.02241; PeerJ. 2018, doi: 10.7717/peerj.4872). In these earlier studies, the authors also provided insights into the mechanisms underlying these effects, which are not addressed in the current study.

2.      The authors should consider providing methods to enhance the antibacterial efficacy of TC against S. mutans or presenting in vivo data to substantiate the effectiveness of TC."

Reviewer 5 Report

Comments and Suggestions for Authors

This research aims to assess the anti-bacterial and anti-biofilm effect of trans-cinnamal-16 dehyde on S. mutans

The topic is interesting and original in the field. The research methodology was applied correctly and carefully. All the references are appropriate.

Some small changes are necessary in order to accept the article for publication:

1) was a power analysis carried out to evaluate the number of samples needed?

2) Authors should include a paragraph explaining the clinical applications of the research.

I believe that with small modification the article could be accepted for publication

Round 2

Reviewer 1 Report

Comments and Suggestions for Authors

agree to accept

Author Response

Thank you for your review. 

We tried to improve the manuscript accordingly. 

Reviewer 4 Report

Comments and Suggestions for Authors

Although the authors have made improvements to the quality of the manuscript, the issue of novelty remains unaddressed. Here are my comments:

1.      The authors should provide a detailed description of the method and standards used to determine the Minimal Bactericidal Concentration (MBC) of 2500 μg/mL using a microplate spectrophotometer.

2.      In the Agar Diffusion Test (ADT), a concentration of 5000 μg/mL shows no bactericidal effects. Despite this, the authors define MBC as 2500 μg/mL. Since concentrations greater than MBC should have a bactericidal effect, the accuracy of MBC=2500 μg/mL should be discussed, especially in comparison to previous studies (Reference 15). The authors should address this in the manuscript.

3.      Regarding Figure 11B, why does the lowest concentration of trans-cinnamaldehyde (TC), i.e., 312.5 and 156.25 μg/mL, induce a dramatic increase in propidium iodide (PI) fluorescence? Furthermore, why do concentrations of 625 and 1250 μg/mL still show a significant increase in PI fluorescence? This phenomenon should be discussed in the manuscript. It is also important to explain why results for concentrations of 2500 and 5000 μg/mL are not presented.

Author Response

  • “Although the authors have made improvements to the quality of the manuscript, the issue of novelty remains unaddressed”

Thank you for this comment. With respect to the novelty, we are the first to show an action on preformed biofilm with TC which is very important for any clinical application. Furthermore, these results set the basis for our subsequent research on caries models using extracted mice jaws. We added the following sentences to the article: “The ability of the TC to affect the biofilm after it was established was proven. These finding set the basis for future in vivo studies.” (Lines 397- 380).

  • “The authors should provide a detailed description of the method and standards used to determine the Minimal Bactericidal Concentration (MBC) of 2500 μg/mL using a microplate spectrophotometer”

In order to determine the MBC, we used the microdilution method with agar plates for MBC measurement which consisted of culturing our UA159 S. mutans with the different TC concentrations every 4 hours for 24 hours and counting the number of colonies formed on the agar plates using the CFU/ml formula. The lowest concentration to kill 99.9% of the bacteria in this case the absence of colonies on the agar plates was chosen as the MBC (2500 our study).

  • In the Agar Diffusion Test (ADT), a concentration of 5000 μg/mL shows no bactericidal effects. Despite this, the authors define MBC as 2500 μg/mL. Since concentrations greater than MBC should have a bactericidal effect, the accuracy of MBC=2500 μg/mL should be discussed, especially in comparison to previous studies (Reference 15). The authors should address this in the manuscript”

Thank you for the remark. The low diffusion and hydrophobic nature of TC were the factors we mentioned to explain the absence of a visible halo on the ADT. The TC doses were placed on sterilized concentric chromatography papers and we believe they did not diffuse through these chromatography papers, hence no visible halos. 

We addressed this issue in the discussion section in lines “385-387”

  • “Regarding Figure 11B, why does the lowest concentration of trans-cinnamaldehyde (TC), i.e., 312.5 and 156.25 μg/mL, induce a dramatic increase in propidium iodide (PI) fluorescence? Furthermore, why do concentrations of 625 and 1250 μg/mL still show a significant increase in PI fluorescence? This phenomenon should be discussed in the manuscript. It is also important to explain why results for concentrations of 2500 and 5000 μg/mL are not presented”

We added the following sentences to the article: “Using 156.25 μg/mL and 312.5μg/mL concentrations resulted in partial inhibition of S. mutans growth, as can be seen in figures 2,3 and 7. Hence, the dramatic increase in the number of dead cells staining with PI was due to an overall higher number of dead cells. When using concentrations 625 μg/mL and 1250 μg/mL, as expected, we can observe more cell death and some albeit less biofilm activity. The 2500 μg/mL and 5000 μg/mL concentrations caused total bacterial kill after 24 hours, thus were not included in the following imaging experiments work due to the fact that these concentrations resulted in almost no live bacteria, causing 99.9% of cell death as ≥ the MBC dose. When thinking of human clinical application, we need to use the lowest Tc concentration possible”. (lines 395-403)

Round 3

Reviewer 4 Report

Comments and Suggestions for Authors

I have no other comments.